# The Global Anchor Method for Quantifying Linguistic Shifts and Domain Adaptation

**Zi Yin**
Department of Electrical Engineering
Stanford University
s09600974@gmail.com

**Vin Sachidananda**
Department of Electrical Engineering
Stanford University
vsachi@stanford.edu

**Balaji Prabhakar**
Department of Electrical Engineering and
Department of Computer Science
Stanford University
balaji@stanford.edu

## Abstract

Language is dynamic, constantly evolving and adapting with respect to time, domain or topic. The adaptability of language is an active research area, where researchers discover social, cultural and domain-specific changes in language using distributional tools such as word embeddings. In this paper, we introduce the global anchor method for detecting corpus-level language shifts. We show both theoretically and empirically that the global anchor method is equivalent to the alignment method, a widely-used method for comparing word embeddings, in terms of detecting corpus-level language shifts. Despite their equivalence in terms of detection abilities, we demonstrate that the global anchor method is superior in terms of applicability as it can compare embeddings of different dimensionalities. Furthermore, the global anchor method has implementation and parallelization advantages. We show that the global anchor method reveals fine structures in the evolution of language and domain adaptation. When combined with the graph Laplacian technique, the global anchor method recovers the evolution trajectory and domain clustering of disparate text corpora.

## 1 Introduction

Linguistic variations are commonly observed among text corpora from different communities or time periods [9, 11]. Domain adaptation seeks to quantify the degree to which language varies in distinct corpora, such as text from different time periods or academic communities such as computer science and physics. This adaptation can be performed either at a word-level–to determine if a particular word's semantics are different in the two corpora, or at the corpus-level–to determine the similarity of language usage in the two corpora. Applications of these methods include identifying how words or phrases differ in meaning in different corpora or how well text-based models trained on one corpus can be transferred to other settings. In this paper, we focus on corpus-level adaptation methods which quantify the structural similarity of two vector space embeddings each learned on a separate corpus.

Consider a motivating example of training conversational intent and entity classifiers for computer software diagnosis. While many pre-trained word embeddings are available for such types of natural language problems, most of these embeddings are trained on general corpora such as news collections or Wikipedia. As previously mentioned, linguistic shifts can result in semantic differences between the domain on which the embeddings were trained and the domain in which the embeddings are

being used. Empirically, such variations can significantly affect the performance of models using embeddings not trained on the target domain, especially when training data is sparse. As a result, it is important, both practically and theoretically, to quantify the domain-dissimilarity in target and source domains as well as study the root cause of this phenomena - language variations in time and domain.

Current distributional approaches for corpus-level adaptation are alignment-based. Consider two corpora $\mathcal{E}$ and $\mathcal{F}$ with corresponding vector embedding matrices $E$ and $F \in \mathbb{R}^{n \times d}$, where $d$ is the dimension of the embeddings and $n$ is the size of the common vocabulary. Using the observation that vector embeddings are equivalent up to a unitary transformation [11, 12, 33], alignment-based approaches find a unitary operator $Q^* = \min_{Q \in O(d)} \|E - FQ\|_F$, where $O(d)$ is the group of $d \times d$ unitary matrices and $\| \ \|_F$ is the Frobenius norm. The shift in the meaning of an individual word can be measured by computing the norm of the difference of the corresponding row in $E$ and $FQ^*$. The difference in language usage between the corpora is then quantified as $\|E - FQ^*\|_F$. In the rest of the paper, all matrix norms will be assumed to be the Frobenius norm unless otherwise specified.

On the other hand, anchor-based approaches [10, 17, 18] are primarily used as a local method for detecting word-level adaptations. In the local anchor method, a set of words appearing in both corpora are picked as "anchors" against which the particular word is compared. If the relative position of the word's embedding to the anchors has shifted significantly between the two embeddings, the meaning of the word is likely to be different. The anchor words are usually hand selected to reflect word meaning shift along a specific direction. For example in Bolukbasi et al. [3], the authors selected gender-related anchors to detect shifts in gender bias. However, the local nature and the need for anchors to be picked by hand or by nearest neighbor search make the local anchoring method unsuitable for detecting corpus-level shifts.

The three major contributions of our work are:

1. Proposing the global anchor method, a generalization of the local anchor method for detecting corpus-level adaptation.

2. Establishing a theoretical equivalence of the alignment and global anchor methods in terms of detection ability of corpus-level language adaptation. Meanwhile, we find that the global anchor method has practical advantages in terms of implementation and applicability.

3. Demonstrating that, when combined with spectral methods, the anchor method is capable of revealing fine details of language evolution and linguistic affinities between disjoint communities.

## 2 Related Work

The study of domain adaptation of natural language, such as diachronic shifts, is an active research field, with word- and corpus-level adaptation constituting the two main topics.

### 2.1 Word-level Adaptation

**Non-Distributional Approaches.** Word-level adaptation methods quantify the semantic and syntactic shift of individual words in different text corpora such as those from disparate communities or time periods. Graph-based methods [5] such as Markov clustering have been used to identify multiple word senses in varying contexts and are useful for resolving ambiguity related to polysemous words, words which have multiple meanings. Topic modeling algorithms, such as the Hierarchal Dirichlet Process (HDP) [19], have also been applied to learn variations in word sense usage. The value of word sense induction methods for understanding word-level adaptation is due to some word senses occurring more or less frequently across different domains (corpora). For instance, consider the word "arms" which can either mean body parts or weapons. A medical corpus may have a higher relative frequency of the former sense when compared to a news corpus. Frequency statistics, which use relative word counts, have been used to predict the rate of lexical replacement in various Indo-European languages over time [26, 28], where more common words are shown to evolve or be replaced at a slower rate than those less frequently used.

**Distributional Approaches.** Distributional methods for word-level shifts use second order statistics, or word co-occurrence distributions, to characterize semantic and syntactic shifts of individual words in different corpora. Distributional methods have been used to determine whether different senses for

a word have been introduced, removed, or split by studying differences in co-occurring words across corpora from disparate time-periods [14, 25]. Vector space embedding models, such as Word2Vec [24], learn vector representations in the Euclidean space. After training embeddings on different corpora, such as Google Books for disjoint time periods, one can compare the nearest neighbors of a particular word in different embedding spaces to detect semantic variations [8, 16, 11, 31]. When the nearest neighbors are different in these embedding spaces for a particular word, it is likely that the meaning of the word is different across the two corpora. The introduction of the anchoring approach extends this idea by selecting the union of a word's nearest neighbors in each embedding space as the set of anchors and is used to detect word-level linguistic shifts due to cultural factors [10]. Anchoring methods have also been used to by compare word embeddings learned from diachronic corpora such as periods of war using a supervised selection of "conflict-specific" anchor words [17, 18].

## 2.2  Corpus-level Adaptation

In contrast to word-level adaptation, corpus-level adaptation methods are used to compute the semantic similarity of natural language corpora. Non-distributional methods such as Jensen Shannon Divergence (JSD), have been applied to count statistics and t-SNE embeddings to study the linguistic variations in the Google Books corpus over time [27, 35].

Alignment-based distributional methods make use of the observation that vector space embeddings are rotation invariant and as a result are equivalent up to a unitary transformation [11, 33]. Alignment methods, which learn a unitary transform between two sets of word embeddings, use the residual loss of the alignment objective to quantify the linguistic dissimilarity between the corpora on which the embeddings were trained. In the context of multi-lingual corpora, Mikolov et al. [23] finds that the alignment method works as well as neural network-based methods for aligning two embedding spaces trained on corpora from different languages. Furthermore, algorithms for jointly training word embeddings from diachronic corpora have been researched to discover and regularize corpus-level shifts due to temporal factors [30, 36]. In the context of diachronic word shifts, Hamilton et al. [11] aligns word embeddings trained on diachronic corpora using the alignment method. In Hamilton et al. [10], anchoring is proposed specifically as a word-level "local" method while alignment is used to capture corpus-level "global" shifts. Similar concepts are used in tensor-based schemes [39, 40] and recommendation systems based on deep-learning [15, 41].

## 3  Global Anchor Method for Detecting Corpus-Level Language Shifts

Given two corpora $\mathcal{E}$ and $\mathcal{F}$, we ask the fundamental question of how different they are in terms of language usage. Various factors contribute to the differences, for example, chronology or community variations. Let $E$, $F$ be two separate word embeddings trained on $\mathcal{E}$ and $\mathcal{F}$ and consisting of common vocabulary. As a recap, the alignment method finds an orthogonal matrix $Q^*$ which minimizes $\|E - FQ\|$, and the residual $\|E - FQ^*\|$ is the dissimilarity between the two corpora.

We propose the global anchor method, a generalization of the local anchor method for detecting corpus level adaptation. We first introduce the local anchor method for word-level adaptation detection, upon which our global method is constructed.

## 3.1  Local Anchor Method for Word-Level Adaptation Detection

The shift of a word's meaning can be revealed by comparing it against a set of anchor words [18, 17], which is a direct result of the distributional hypothesis [10, 13, 6]. Specifically, let $\{1, \cdots, l\}$ be the indices of the $l$ anchor words, common to two different corpora. To measure how much the meaning of a word $i$ has shifted between the two corpora, one triangulates it against the $l$ anchors in the two embedding spaces by calculating the inner products of word $i$'s vector representation with those of the $l$ anchors. Since the embedding for word $i$ is the $i$-th row of the embedding matrix, this procedure produces two length-$l$ vectors, namely

$$(\langle E_{i,\cdot}, E_{1,\cdot}\rangle, \cdots, \langle E_{i,\cdot}, E_{l,\cdot}\rangle) \text{ and } (\langle F_{i,\cdot}, F_{1,\cdot}\rangle, \cdots, \langle F_{i,\cdot}, F_{l,\cdot}\rangle).$$

The norm of the difference of these two vectors,

$$\|\langle E_{i,\cdot}, E_{1,\cdot}\rangle, \cdots, \langle E_{i,\cdot}, E_{l,\cdot}\rangle) - (\langle F_{i,\cdot}, F_{1,\cdot}\rangle, \cdots, \langle F_{i,\cdot}, F_{l,\cdot}\rangle)\|$$

reflects the drift of word $w_i$ with respect to the $l$ anchor words. The anchors are usually selected as a set of pre-defined words in a supervised fashion or by a nearest neighbor search, to reflect shifts along a specific direction [3, 17] or a local neighborhood [10].

## 3.2 The Global Anchor Method

Two issues arise from the local anchor method for corpus-level adaptation, namely its local nature and the need of anchors to be hand-picked or selected using nearest neighbors. We address them by introducing the global anchor method, a generalization of the local approach. In the global anchor method, we use all the words in the common vocabulary as anchors, which brings two benefits. First, human supervision is no longer needed as anchors are no longer hand picked. Second, the anchor set is enriched so that shift detections are no longer restricted to one direction. These two benefits make the global anchor method suitable for detecting corpus level adaptation. In the global anchor method, the expression for the corpus-level dissimilarity simplifies to:

$$\|EE^T - FF^T\|.$$

Consider the $i$-th row of $EE^T$ and $FF^T$ respectively. $(EE^T)_{i,\cdot} = (\langle E_{i,\cdot}, E_{1,\cdot}\rangle, \cdots, \langle E_{i,\cdot}, E_{n,\cdot}\rangle)$ which measures the $i$-th vector $E_i$ using all other vectors as anchors. The same is true for $(FF^T)_{i,\cdot}$. The norm of the difference, $\|(EE^T)_{i,\cdot} - (FF^T)_{i,\cdot}\|$, measures the relative shift of word $i$ across the two embeddings. If this distance is large, it is likely that the meaning of the $i$-th word is different in the two corpora. This leads to an embedding distance metric also known as the Pairwise Inner Product loss [37, 38].

# 4 The Alignment and Global Anchor Methods: Equivalent Detection of Linguistic Shifts

Both the alignment and global anchor methods provide metrics for corpus dissimilarity. We prove in this section that the metrics which the two methods produce are equivalent. The proof is based on the isotropy observation of vector embeddings [1] and projection geometry. Recall from real analysis [29], that two metrics $d_1$ and $d_2$ are equivalent if there exist positive $c_1$ and $c_2$ such that:

$$c_1 d_1(x, y) \le d_2(x, y) \le c_2 d_1(x, y), \ \forall x, y.$$

## 4.1 The Isotropy of Word Vectors

We show that the columns of embedding matrices are approximately orthonormal, which arises naturally from the isotropy of word embeddings [1]. The isotropy requires the distribution of the vectors to be uniform along all directions. This implies $E_w/\|E_w\|$ follows a uniform distribution on a sphere, which is equivalent in distribution to the case when $E_w$ has i.i.d., zero-mean normal entries [21]. Under this assumption, we invoke a result by Bai and Yin [2]:

**Theorem 1.** *Suppose the entries of $X \in \mathbb{R}^{n \times d}$, $d/n = p \in (0, 1)$, are random i.i.d. with zero mean, unit variance, and finite $4^{th}$ moment. Let $\lambda_{\min}$ and $\lambda_{\max}$ be the smallest and largest singular values of $X^T X/n$, respectively. Then:*

$$\lim_{n \to \infty} \lambda_{\min} \overset{a.c.}{=} (1 - \sqrt{p})^2, \ \lim_{n \to \infty} \lambda_{\max} \overset{a.c.}{=} (1 + \sqrt{p})^2$$

This shows that the largest and smallest singular values for the embedding matrix, under the i.i.d. assumption, are asymptotically $\sqrt{n}\sigma(1 - \sqrt{d/n})$ and $\sqrt{n}\sigma(1 + \sqrt{d/n})$ respectively. Further, notice the dimensionality $d$, usually in the order of hundreds, is much smaller than the vocabulary size $n$, which can be tens of thousands up to millions [24]. This leads to the result that the singular values of the embedding matrix should be tightly clustered, which is empirically verified by Arora et al. [1]. In other words, the columns of $E$ and $F$ are close to orthonormal.

## 4.2 The Equivalence of the Global Anchor Method and the Alignment Method

The orthonormality of the columns of $E$ and $F$ means they can be viewed as the basis for the subspaces they span. Lemma 2 is a classical result regarding the principal angles [7] between subspaces. Using the lemma, we prove Theorems 3 and 4, and Corollary 4.1.

**Lemma 2.** *Suppose $E \in \mathbb{R}^{n \times d}$, $F \in \mathbb{R}^{n \times d}$ are two matrices with orthonormal columns. Then:*

1. *SVD($E^T F$)= $UCV^T$, where $C_i = \cos(\theta_i)$ is the cosine of the $i^{th}$ principal angle between subspaces spanned by the columns of $E$ and $F$.*

2. *SVD($E_{\perp}^T F$)= $\tilde{U}SV^T$, where $S_i = \sin(\theta_i)$ is the sine of the $i^{th}$ principal angle between subspaces spanned by the columns of $E$ and $F$, where $E_{\perp} \in \mathbb{R}^{n \times (n-d)}$ is an orthogonal basis for $E$'s null space.*

Let $\Theta = (\theta_1, \cdots, \theta_d)$ be the vector of principal angles between the subspaces spanned by $E$ and $F$, and all operations on $\Theta$, such as $\sin$ and raising to a power, be applied element-wise.

**Theorem 3.** *The metric for the alignment method, $\|E - FQ^*\|$, equals $2\|\sin(\Theta/2)\|$.*

*Proof.* Note that

$$(E - FQ)(E - FQ)^T = EE^T + FF^T - EQ^T F^T - FQE^T$$

We perform a change of basis into the columns of $[E \; E_{\perp}]$,

$$(E - FQ)(E - FQ)^T$$
$$= [E \quad E_{\perp}] \left( \begin{bmatrix} I & 0 \\ 0 & 0 \end{bmatrix} + \begin{bmatrix} E^T FF^T E & E^T FF^T E_{\perp} \\ E_{\perp}^T FF^T E & E_{\perp}^T FF^T E_{\perp} \end{bmatrix} - \begin{bmatrix} E^T FQ & 0 \\ E_{\perp}^T FQ & 0 \end{bmatrix} - \begin{bmatrix} Q^T F^T E & Q^T F^T E_{\perp} \\ 0 & 0 \end{bmatrix} \right) \begin{bmatrix} E^T \\ E_{\perp}^T \end{bmatrix}$$
$$= [E \quad E_{\perp}] \left( \begin{bmatrix} I & 0 \\ 0 & 0 \end{bmatrix} + \begin{bmatrix} UC^2U^T & UCS\tilde{U}^T \\ \tilde{U}CSU^T & \tilde{U}S^2\tilde{U}^T \end{bmatrix} - \begin{bmatrix} UCV^TQ & 0 \\ \tilde{U}SV^TQ & 0 \end{bmatrix} - \begin{bmatrix} Q^TVCU^T & Q^TVS\tilde{U}^T \\ 0 & 0 \end{bmatrix} \right) \begin{bmatrix} E^T \\ E_{\perp}^T \end{bmatrix} \tag{1}$$

Notice that the $Q^*$ minimizing $\|E - FQ^*\|$ equals $Q^* = VU^T$ [32]. Plug in $Q^*$ to (1) and we get

$$(E - FQ)(E - FQ)^T = [E \quad E_{\perp}] \begin{bmatrix} U & 0 \\ 0 & \tilde{U} \end{bmatrix} \begin{bmatrix} I + C^2 - 2C & CS - S \\ CS - S & S^2 \end{bmatrix} \begin{bmatrix} U & 0 \\ 0 & \tilde{U} \end{bmatrix}^T \begin{bmatrix} E^T \\ E_{\perp}^T \end{bmatrix}$$
$$= [E \quad E_{\perp}] \begin{bmatrix} U & 0 \\ 0 & \tilde{U} \end{bmatrix} \begin{bmatrix} (I - C)^2 & -S(1 - C) \\ -S(1 - C) & S^2 \end{bmatrix} \begin{bmatrix} U & 0 \\ 0 & \tilde{U} \end{bmatrix}^T \begin{bmatrix} E^T \\ E_{\perp}^T \end{bmatrix} \tag{2}$$

By applying the trigonometric identities $1 - \cos(\theta) = 2\sin^2(\theta/2)$ and $\sin(\theta) = 2\sin(\theta/2)\cos(\theta/2)$ to equation (2), we have

$$(I - C)^2 = 4\sin^4(\Theta/2), \quad -S(1 - C) = -4\sin^3(\Theta/2)\cos(\Theta/2), \quad S^2 = 4\sin^2(\Theta/2)\cos^2(\Theta/2)$$

Plug in the quantities into (2),

$$= [E \quad E_{\perp}] \begin{bmatrix} U & 0 \\ 0 & \tilde{U} \end{bmatrix} \begin{bmatrix} \sin(\Theta/2) \\ -\cos(\Theta/2) \end{bmatrix} 4\sin^2(\Theta/2) \begin{bmatrix} \sin(\Theta/2) \\ -\cos(\Theta/2) \end{bmatrix}^T \begin{bmatrix} U & 0 \\ 0 & \tilde{U} \end{bmatrix}^T \begin{bmatrix} E^T \\ E_{\perp}^T \end{bmatrix}$$

As a result, the singular values of $E - FQ^*$ are $2\sin(\Theta/2)$. So $\|E - FQ^*\| = 2\|\sin(\Theta/2)\|$ $\quad\square$

**Theorem 4.** *The metric for the global anchor method, $\|EE^T - FF^T\|$, equals $\sqrt{2}\|\sin\Theta\|$.*

*Proof.* First, notice $[E \; E_{\perp}]$ forms a unitary matrix of $\mathbb{R}^n$. Also note the Frobenius norm is unitary-invariant. The above observations allow us to perform a change of basis:

$$\|EE^T - FF^T\| = \left\| \begin{bmatrix} E^T \\ E_{\perp}^T \end{bmatrix} (EE^T - FF^T) [E \quad E_{\perp}] \right\| = \left\| \begin{bmatrix} I & 0 \\ 0 & 0 \end{bmatrix} - \begin{bmatrix} E^T FF^T E & E^T FF^T E_{\perp} \\ E_{\perp}^T FF^T E & E_{\perp}^T FF^T E_{\perp} \end{bmatrix} \right\|$$
$$= \left\| \begin{bmatrix} I & 0 \\ 0 & 0 \end{bmatrix} - \begin{bmatrix} UC^2U^T & UCS\tilde{U}^T \\ \tilde{U}CSU^T & \tilde{U}S^2\tilde{U}^T \end{bmatrix} \right\| = \left\| \begin{bmatrix} U & 0 \\ 0 & \tilde{U} \end{bmatrix} \begin{bmatrix} I - C^2 & -CS \\ -CS & -S^2 \end{bmatrix} \begin{bmatrix} U & 0 \\ 0 & \tilde{U} \end{bmatrix}^T \right\|$$
$$= \left\| \begin{bmatrix} S^2 & -CS \\ -CS & -S^2 \end{bmatrix} \right\| = \left\| \begin{bmatrix} S & 0 \\ 0 & S \end{bmatrix} \begin{bmatrix} S & -C \\ -C & -S \end{bmatrix} \right\| = \left\| \begin{bmatrix} S & 0 \\ 0 & S \end{bmatrix} \right\|$$
$$= \sqrt{2}\|S\| = \sqrt{2}\|\sin\Theta\|$$

$\square$

**Corollary 4.1.** $\|E - FQ^*\| \leq \|EE^T - FF^T\| \leq \sqrt{2}\|E - FQ^*\|$.

*Proof.* By Theorem 3 and 4, $\|EE^T - FF^T\| = \sqrt{2}\|\sin\Theta\|$ and $\min_{Q \in O(d)} \|E - FQ\| = 2\|\sin(\Theta/2)\|$. Finally, the corollary can be obtained since $\sqrt{2}\sin(\theta/2) \leq \sin(\theta) \leq 2\sin(\theta/2)$, which is a result of $\sqrt{2} \leq 2\cos(\theta/2) \leq 2$ for $\theta \in [0, \pi/2]$. $\qquad\square$

### 4.3 Validation of the Equivalence between Alignment and Global Anchor Methods

We proved that the anchor and alignment methods are equivalent in detecting linguistic variations for two corpora up to at most a constant factor of $\sqrt{2}/2$ under the isotropy assumption. To empirically verify that the equivalence holds, we conduct the following experiment. Let $E^{(i)}$ and $E^{(j)}$ correspond to word embeddings trained on the Google Books dataset for distinct years $i$ and $j$ in $\{1900, 1901, \cdots, 2000\}$ respectively[1]. Normalize $E^{(i)}$ and $E^{(j)}$ by their average column-wise norm, $\frac{1}{d}\sum_{k=1}^{d}\|E_{[\cdot,k]}\|$, so the embedding matrices have the same Frobenius norm. For every such pair $(i, j)$, we compute

$$\frac{\min_{Q \in O(d)} \|E^{(i)} - E^{(j)}Q\|}{\|E^{(i)}E^{(i)^T} - E^{(j)}E^{(j)^T}\|}.$$

Our theoretical analysis showed that this number is between $\sqrt{2}/2 \approx 0.707$ and 1. Since we evaluate for every possible pair of years, there are in total 10,000 such ratios. The statistics are summarized in Table 1. The empirical results indeed match the theoretical analysis. Not only are the ratios within the range $[\sqrt{2}/2, 1]$, but also they are tightly clustered around 0.83 meaning that empirically the output of alignment method is approximately a constant of the global anchor method.

Table 1: The Ratio of Distances Given by the Alignment and Global Anchor Methods

|  | mean | std | min. | max. | theo. min. | theo. max. |
|---|---|---|---|---|---|---|
| Ratio | 0.826 | 0.015 | 0.774 | 0.855 | $\sqrt{2}/2 \approx 0.707$ | 1 |

### 4.4 Advantages of the Global Anchor Method over the Alignment Method

Theorems 3, 4 and Corollary 4.1 together establish the equivalence of the anchor and alignment methods in identifying corpus-level language shifts using word embeddings; the methods differ by at most a constant factor of $\sqrt{2}$. Despite the theoretical equivalence, there are several practical differences to consider. We briefly discuss some of these differences.

- **Applicability:** The alignment methods can be applied only to embeddings of the same dimensionality, since the orthogonal transformation it uses is an isometry onto the original $\mathbb{R}^d$ space. On the other hand, the global anchor method can be used to compare embeddings of different dimensionalities.

- **Implementation:** The global anchor method involves only matrix multiplications, making it convenient to code. It is also easy to parallelize, as the matrix product and entry-wise differences are naturally parameterizable and a map-reduce implementation is straightforward. The alignment method, on the other hand, requires solving a constrained optimization problem. This problem can be solved using gradient [36] or SVD [11] methods.

Due to the benefits in implementation and applicability, along with the established equivalence of the global anchoring and alignment methods, the global anchor method should be preferred for quantifying corpus-level linguistic shifts. In the next section, we conduct experiments using the global anchor method to detect linguistic shifts in corpora which vary in time and domain.

## 5 Detecting Linguistic Shifts with the Global Anchor Method

The global anchor method can be used to discover language evolution and domain specific language shifts. In the first experiment, we demonstrate that the global anchor method discovers the fine-grained evolutionary trajectory of language, using the Google Books N-gram data [20]. The Google Books dataset is a collection of digitized publications between 1800 and 2009, which makes up roughly 6% of the total number of books ever published. The data is presented in $n$-gram format,

where $n$ ranges from 1 to 5. We collected the $n$-gram data for English fictions between 1900 and 2000, trained skip-gram Word2Vec models for each year, and compared the distance between embeddings of different years using the global anchor method.

In our second experiment, we show that the global anchor method can be used to find community-based linguistic affinities of text from varying academic fields and categories on arXiv, an online pre-print service widely used in the fields of computer science, mathematics, and physics. We collected all available arXiv LaTeX files submitted to 50 different academic communities between January 2007 and December 2017 resulting in corpora assembled from approximately 75,000 academic papers, each associated with a single primary academic field and category. After parsing these LaTeX files into natural language, we constructed disjoint corpora and trained skip-gram Word2Vec models for each category. We then compute the anchor loss for each pair of categories.

We conduct two more experiments on Reddit community (subreddit) comments as well as the Corpus of Historical American English (COHA); these experiments along with further analysis on word-level linguistic shifts are deferred to the Appendix due to space constraints. Our codes and datasets are publicly available on GitHub[2].

## 5.1 Language Evolution

The global anchor method can reveal the evolution of language, since it provides a quantitative metric $\|EE^T - FF^T\|$ between corpora from different years. Figure 1 is a visualization of the anchor distance between different embeddings trained on the Google Books n-gram dataset, where the $ij^{th}$ entry is the anchor distance between $E^{(i)}$ and $E^{(j)}$. In Figure 1a, we grouped n-gram counts and trained embeddings for every decade, and in Figure 1b the embeddings are trained for every year.

First, we observe that there is a banded structure. Linguistic variation increases with respect to $|i - j|$, which is expected. The banded structure was also observed by Pechenick et al. [27] who used word frequency methods instead of distributional approaches. Languages do not evolve at constant speed and the results of major events can have deep effects on the evolution of natural language. Due to the finer structure of word embeddings, compared to first-order statistics like frequencies, the anchor method captures more than just banded structure. An example is the effect of wars. In Figure 1b, we see that embeddings trained on years between 1940-1945 have greater anchor loss when compared to embeddings from 1920-1925 than those from 1915-1918. Figure 2 demonstrates the row vector in Figure 1b for the year 1944.

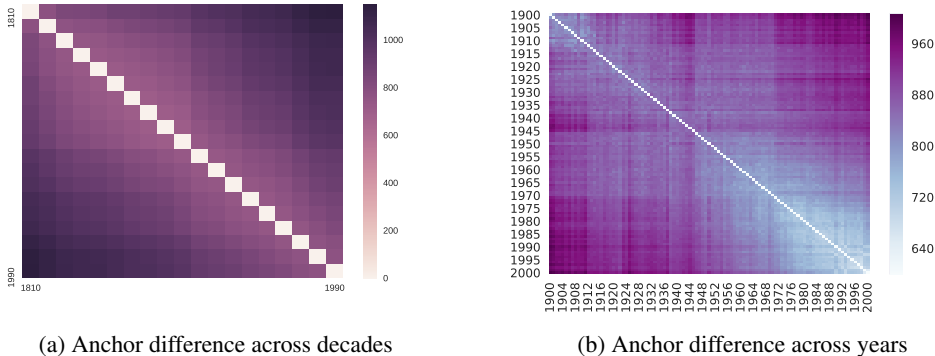

(a) Anchor difference across decades

(b) Anchor difference across years

Figure 1: Temporal evolution of English language and the banded structure

In Figure 2, there is a clear upward trend of the anchor difference as one moves away from 1944. However, there is a major dip around 1915-1918 (WWI), and two minor dips around 1959 (Korean War) and 1967 (Vietnam War). This pattern is consistent across 1939-1945 (WWII) for the anchor methods, but not as clear when using frequency methods. As per the distributional hypothesis [6], one should consider that frequency methods, unlike co-occurrence approaches, do not capture the semantics of words but rather the relative frequency of their usage. As discussed in Pechenick et al.

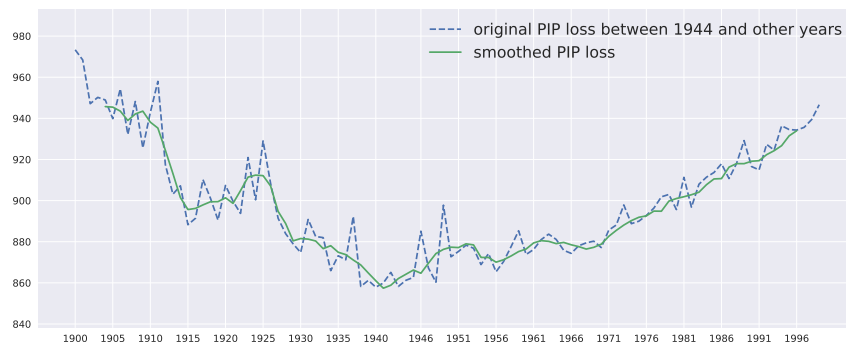

Figure 2: Anchor difference for year 1944, note the dips during war times

[27], frequency change of popular words (his, her, which, *etc.*) contribute the most to the frequency discrepancies. This, however, does not mean the two corpora are linguistically different, as these popular words may retain their meaning and could be used in the same contexts, despite frequency differences. The global anchor method is less sensitive to this type of artifact as it captures change of word meaning rather than frequency, and as a result is able to show finer structures of language shifts.

## 5.2 Trajectory of Language Evolution

As discussed in Section 5.1, the global anchor method can provide finer structure about the rate of evolution compared to frequency-based approaches. The distance matrix provided by the anchor method can further give information about the *direction of evolution* via the graph Laplacian technique [34]. The graph Laplacian method looks for points in a low dimensional space where the distance between the pair $(i, j)$ reflects the corresponding entry of the anchor loss matrix. Algorithm 1 describes the procedure for obtaining Laplacian Embeddings from the anchor loss matrix.

---

**Algorithm 1** Laplacian Embedding for Distance Matrix

---

1: Given a distance matrix $M$
2: Let $S = \exp\left(-\frac{1}{2\sigma^2} M\right)$ be the exponentiated similarity matrix;
3: Calculate the Laplacian $L = I - D^{-1/2} S D^{-1/2}$, where $D = \text{diag}(d)$ and $d_i = \sum_j S_{ij}$;
4: Compute the singular value decomposition $UDV^T = L$;
5: Take the last $k$ columns of $U$, $U_{\cdot, n-k:n}$, as the dimension $k$ embedding of $M$.

---

In Figure 3a, we show the 2-dimensional embedding of the anchor distance matrix for Google Books n-gram (English Fiction) embeddings from year 1900 to 2000. We can see that the years follow a trajectory starting from the bottom-left and gradually ending at the top-right. There are a few noticeable deviations on this trajectory, specifically the years 1914-1918, 1938-1945 and 1981-2000. It is clear that the first two periods were major war-times, and these two deviations closely resemble each other, indicating that are driven by the same type of event. The last deviation is due to the rise of scientific literature, where a significant amount of technical terminologies (*e.g.* computer) were introduced starting from the 1980s. This was identified as a major bias of Google Books dataset [27].

## 5.3 Linguistic Variation in Academic Subjects

In Figure 3b, we use the global anchor method to detect linguistic similarity of arXiv papers from different academic communities. We downloaded and parsed the LaTeX files posted on arXiv between Jan. 2007 and Dec. 2017, and trained embeddings for each academic category using text from the corresponding papers. The anchor distance matrix is deferred to the appendix due to page limits. We applied the Laplacian Embedding, Algorithm 1, to the anchor distance matrix, and obtained spectral embeddings for different categories. It can be observed that the categories are generally clustered according to their fields; math, physics and computer science categories all forms their own clusters. Additionally, the global anchor method revealed a few exceptions which make sense at second glance:

- Statistical Mechanics (cond-mat.stat-mech) is closer to math and computer science categories
- History and Overview of Mathematics (math.HO) is far away from other math categories
- Information theory (cs.IT) is closer to math topics than other computer science categories

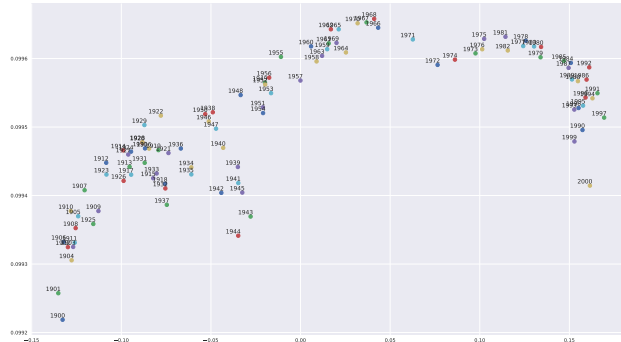

(a) Anchor difference across years of N-gram corpus

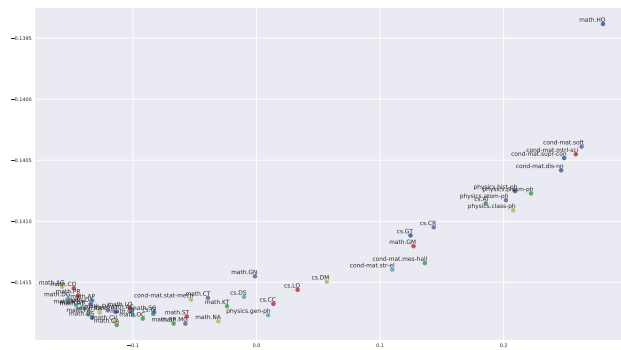

(b) Anchor difference across ArXiv topics

Figure 3: 2-D embedding of corpora reveals evolution trajectory and domain similarity

## 6 Conclusion and Future Work

In this paper, we introduced the global anchor method for detecting corpus-level linguistic shifts. We showed both theoretically and empirically that the global anchor method provides an equivalent metric to the alignment method, a widely used method for corpus-level shift detection. Meanwhile, the global anchor method excels in applicability and implementation. We demonstrated that the global anchor method can be used to capture linguistic shifts caused by time and domain. It is able to reveal finer structures compared to frequency-based approaches, such as linguistic variations caused by wars and linguistic similarities between academic communities.

We demonstrated in Section 5 important applications of the global anchor method in detecting diachronic and domain-specific linguistic shifts using word embeddings. As embedding models are foundational tools in Deep Learning, the global anchor method can be used to address the problems of Transfer Learning and Domain Adaptation, which are ubiquitous in NLP and Information Retrieval. In these fields, Transfer Learning is important as it attempts to use models learned from a source domain effectively in different target domains, potentially with much smaller amounts of data. The efficacy of model transfer depends critically on the domain dissimilarity, which is what our method quantifies.

While we mainly discuss corpus-level adaptation in this paper, future work includes using the anchor method to discover global trends and patterns in different corpora, which lies between corpus and word-level linguistic shifts. In particular, unsupervised methods for selecting anchors are of great interest.

# 7 Acknowledgements

The authors would like to thank Professors Dan Jurafsky and Will Hamilton for their helpful comments and discussions. Additionally, we thank the anonymous reviewers for their feedback during the review process.

## Footnotes

[1]Detail of the dataset and training will be discussed in the Section 5.

[2]https://github.com/ziyin-dl/global-anchor-method

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
