[Supplementary Material]

# 8 Appendix

## 8.1 Calculation of Alignment Operator: Orthogonal Procrustes Problem

The objective of the aligment method in Section 3.1 is the Orthogonal Procrustes Problem [32] which has an analytic solution. Let $SVD(E^T F) = U\Sigma V^T$:

$$
\begin{aligned}
Q^* &= \arg \min_{Q \in O(d)} \|E - FQ\|^2 \\
&= \arg \min_{Q \in O(d)} \|E\|^2 + \|FQ\|^2 - 2\langle E, FQ \rangle \\
&= \arg \max_{Q \in O(d)} \langle E^T F, Q \rangle \\
&= VU^T
\end{aligned}
$$

Using the above analytic solution, we can calculate an alignment operator for a pair of word embeddings. For cases where performing SVD is not computationally feasible, gradient descent with projection onto the orthogonal group may be performed [36].

## 8.2 Effect of Embedding Dimensionality on Equivalence Constant

We conducted more experiments to study the effect of dimensionality on the equivalence relation between the global anchor method and the alignment method. The results are summarized in Table 2. We found the constant factor is always around 0.8 with little variation, regardless of the dimensions. The Google Books and arXiv experiments used embeddings of dimensions 300 and 50, respectively.

Table 2: Ratio between Distance by Alignment and Anchor Methods

|                  | dimension | mean  | std   | min.  | max.  |
|------------------|-----------|-------|-------|-------|-------|
| alignment/anchor | 100       | 0.797 | 0.018 | 0.754 | 0.821 |
| alignment/anchor | 200       | 0.811 | 0.015 | 0.779 | 0.838 |
| alignment/anchor | 400       | 0.814 | 0.019 | 0.753 | 0.849 |
| alignment/anchor | 500       | 0.809 | 0.024 | 0.737 | 0.847 |

## 8.3 Construction of Vector Space Embeddings

In constructing our word embeddings on corpora from our experiments, we use the Word2Vec model class in the Gensim Python package with hyperparameters similar to those in [24].

- **Google Books**: Skip-gram with Negative Sampling; Dimensionality: 300, Window size: 5, Minimum word frequency: 100, Negative samples: 4, Iterations: 15 epochs.

- **arXiv:** Skip-gram with Negative Sampling; Dimensionsionality: 50, Window size: 5 , Minimum word frequency: 3, Negative samples: 3, Iterations: 5 epochs, Random seed: 1.

- **Reddit:** (Same as arXiv)

- **COHA:** Skip-gram with Negative Sampling; Dimensionality: 100, Window size: 5, Minimum word frequency: 10, Negative samples: 4, Iterations: 15 epochs.

## 8.4 Corpora Statistics

We provide some high level statistics of the corpora used in our two experiments.

- **Google Books**: Number of n-grams range from a few million to a few tens of millions depending on the year. Number of years: 101, Size of anchoring set (common vocabulary): 13,000.

- **arXiv:** Number of categories: 50, Category average corpus size (in words): 10,932,027, Category average number of unique tokens: 32,572, Size of anchoring set (common vocabulary): 2,969

- **Reddit:** Number of categories: 50, Category average corpus size (in words): 11,487,859, Category average number of unique tokens: 81,717, Size of anchoring set (common vocabulary): 10,158

- **COHA:** Number of n-grams range from 14 million to 20 million depending on the decade, for more information please see COHA corpora sizes. Number of decades: 18 (1810 and 1820 corpora omitted), Size of anchoring set (common vocabulary): 7,973.

## 8.5 Further arXiv Experimentation

In this section, we include the anchoring loss matrix for the arXiv academic corpora as well as word-level linguistic variations in and between various academic fields.

### 8.5.1 Anchoring Loss Matrix for arXiv Experiments

Figure 4 shows the anchoring loss for pairs of 50 different academic categories from the fields of computer science, mathematics, and physics on arXiv. Categories within the same field are grouped together in the figure since the ordering of the categories is alphabetical. As expected, we notice a block structure which signifies that categories are typically most similar to others in the same field.

Figure 4: Anchor Difference across arXiv categories

Through further observation of the anchoring distance matrix, we find interesting linguistic patterns associated with academic writing in the various communities:

- Among the different fields, academic writing is most similar across different categories within the field of Mathematics (MA). We find that words commonly used in mathematical proofs, such as 'recall', 'suppose', 'contradiction', 'particular', 'assume', are used in very similar contexts across the different MA categories as measured by the anchoring losses for these individual words. This observation does not hold when comparing the anchoring loss of these words in different fields such as Physics or Computer Science. Given the rigorous structure of language used in mathematical proofs, we find this observation matches our intuition.

- Certain categories in the field of Computer Science (CS), such as Information Theory (cs.IT), uses language which is more similar to MA categories than other CS categories as measured by the anchoring metric. Data structures and Algorithms (cs.DS) and Computational Complexity (cs.CC) are examples of other CS categories whose academic writing more closely resembles language used in MA categories. Interestingly, these three categories are often associated with Theoretical Computer Science, a Computer Science subfield which has historically had strong ties to Mathematical communities.

- Academic language in the Statistical Mechanics (cond-mat.stat-mech) category is more similar to communities in MA and CS than other categories in the field of Condensed Matter Physics (cond-mat). This observation can likely be attributed to the use of Statistical Mechanics in many mathematical settings such as uncertainty propagation and potential games as well computer science settings such as theoretical understanding of Neural Networks dynamics.

- Categories such as History and Overview of Mathematics (math.HO) and General Mathematics (math.GM) are anomalous in that the anchoring loss of these communities is large when compared to embeddings corresponding to almost all other categories. After further consideration, we find that the History and Overview of Mathematics category is less focused on disseminating novel technical results than the other categories used in the experiments. Rather, the History of Mathematics category focuses on biographies, philosophy of mathematics, mathematics education, recreational mathematics, and communication of mathematics. General Mathematics is a placeholder category which is used when authors cannot find an appropriate MA category. Since General Mathematics focuses on topics not covered elsewhere, it makes sense that the language used in the category might be unrelated to the other MA categories.

### 8.5.2 Word-Level linguistic varations between arXiv communities

In Table 3, we show the words with the lowest anchor difference when aggregated over all pairs of subjects within the same academic field. These words are used most similarly across all subjects within the respective field. Note that we do not include stopwords, as defined in the NLTK library, in this analysis.

Table 3: Word with most similar usage in a particular academic field

| Cond-Mat | CS | Math | Physics |
|---|---|---|---|
| normal | integrals | resolving | entries |
| power | circular | spread | triple |
| black | mixing | virtually | dimensional |
| study | notes | contract | relation |
| letters | tail | soft | direct |
| contract | print | backward | quasi |
| sharing | rigid | design | convert |
| notes | post | status | rigid |
| polynomials | rough | shaped | functional |
| plain | standard | maximally | hidden |

### 8.6 COHA Experiments

We conduct experiments on the Corpus of Historical American English (COHA) dataset [4] which contains balanced fiction and non-fiction texts from 1810 to 2009. We downloaded four-gram corpora, grouped and trained embeddings by decade. Texts from the decades following 1810 and 1820 were omitted as these corpora were significantly smaller than others present in the dataset. The anchor distance matrix and Laplacian Embeddings were subsequently calculated on these embeddings.

### 8.6.1 Anchoring Loss Matrix for COHA Experiments

Figure 5 shows the anchoring loss for pairs of the 18 decades used from the COHA dataset. As seen previously in the Google Books dataset, we notice a banded structure where anchor loss monotically increases as we move away from the diagonal - signifying the evolution of language over time.

### 8.6.2 Laplacian Embedding for COHA Experiments

In Figure 6, we use the global anchor method to detect linguistic similarity of text from different decades within the COHA dataset. We applied the Laplacian Embedding, Algorithm 1, to the anchor distance matrix, and obtained spectral embeddings for different decades.

### 8.7 Reddit Experiments

We conduct experiments on all 1.7 billion publicly available Reddit comments from October 2007 to May 2015 [22]. These comments were grouped by community or subreddit to find community-based linguistic affinities. We downloaded the dataset and trained embeddings for the 50 subreddits with the largest corpus size. We then calculate the anchor distance matrix and the Laplacian Embedding.

Figure 5: Anchor Difference across decades (COHA)

Figure 6: Laplacian Embedding of Decades (COHA)

### 8.7.1 Anchoring Loss Matrix for Reddit Experiments

Figure 7 shows the anchoring loss for pairs of the 50 largest subreddits by corpora size. Due to the absence of hierarchy in subreddit structure, we notice a lack of discernible grouping in the distance matrix.

Through further observation of the anchoring distance matrix, we find interesting patterns associated with language used in various subreddits comments:

- Noticeably, the embeddings generated from the AskReddit subreddit corpus have high anchoring loss when compared with embeddings for most other subreddits. After inspection, we notice that the purpose of AskReddit is general question answering compared to most other subreddits which focus on reactionary conversation. Additionally, AskReddit contains a much broader range of discussion topics compared to the directed conversation of other subreddits.

- We find that among the numerous subreddits devoted to discussion of particular games, further distinction can be observed from the anchoring loss matrix based on characteristics of the games themselves. For instance, the subreddit with language most similar to the magicTCG subreddit, as measured by anchoring loss, is hearthstone. Interestingly, these are the only two communities we analyzed that are dedicated to card games. On the other

Figure 7: Anchor Difference across subreddits

hand, embeddings from subreddits dedicated to discussion of role-playing games such as DestinyTheGame and leagueoflegends have low anchoring loss. By inspecting the anchoring loss matrix, we can find variations in language across subreddits dedicated to varying game genres.

- Additionally, we notice that communities focused on sharing or discussing primarily image or video rather than text posts can be revealed through the anchoring metric. In particular, the embeddings constructed from the videos, pics, and movies subreddits have low anchoring loss. This is potentially due to differences in the language used when discussing image or video content compared to text content. Furthermore, subreddits dedicated to memes, such as AdviceAnimals or funny, have similar language usage as measured by anchoring loss.

- While subreddits inherently lack hierarchy, we can use anchoring loss to find communities with similar themes. In particular, we find that the CFB, hockey, nba, soccer, and Squared-Circle subreddits all exhibit similar language usage as measured by anchoring loss. These subreddits make up the five of the six most popular sports communities on Reddit and discuss College Football, Hockey, Professional Basketball, Soccer, and Wrestling, respectively.

### 8.7.2 Laplacian Embedding for Reddit Experiments

In Figure 8, we use the global anchor method to detect linguistic similarity of Reddit comments from different subreddits. We applied the Laplacian Embedding, Algorithm 1, to the anchor distance matrix, and obtained spectral embeddings for different subreddits.

Figure 8: Laplacian Embedding of subreddits