[Reviews · NeurIPS 2018]

Reviewer 1



This paper is about methods for comparing embedding spaces. The paper suggests a extension of a local technique that looks at just the differences in distances from a few words (rows) to a global generalization that considers the differences between all words. The paper then theoretically shows that this metric is equivalent (and within a small constant) of an alignment method, which is widely used but more expensive to compute. The proposed method is then used to compare embeddings across years of the Google book corpus. The distance between each pair of years falls within the small constant that was proven (an empirical sanity check) and the distances over time correspond well to real word events. The writing is very clear, and the paper is well-organized. The proofs are clean and compact (although I admit I did not check all the math). A figure or diagram might improve the exposition of the methods section. The paper includes both theoretical and empirical results. Besides being algorithmically simpler than the alignment approach, the global anchors approach allows comparisons of embeddings of different dimensionality. The proposed application is identifying linguistic shifts over time, which is somewhat narrowly defined. However others have used related methods for aligning word embeddings cross-lingually, so this may have broader implications. I have read the author response and my review above still stands.

Reviewer 2



In this work authors introduce a method for quantifying linguistics shifts on a corpus level which they refer to as the global anchor method (GAM). This method is based on the already proposed local anchor method (LAM). Unlike LAM which is used for adaptation detection on word level, the GAM generalizes to corpus level adaptation. Unlike LAM where anchor words are selected using certain heuristics or NN search in the GAM all common words across the two collections are used as anchor words. In addition GAM also allows for embeddings of different dimensions to be compared which is not possible with the alignment method. Lastly, unlike LAM, GAM is more optimized and scalable to compute as it uses only matrix multiplication. Authors also theoretically show that the GAM and the existing alignment method are equivalent. Overall I think that the work presented in this paper is interesting and the presentation follows a relatively concise and clear path. On the other hand it's a small addition to a very narrow domain. As such I think the the work presented here may be more suitable for an NLP conference or a workshop. It would be good to perform the empirical analysis of the constant factor relationship using embeddings of different dimensions. It would also be interesting to see a plot as to how the ration changes with the number of dimensions. What embedding dimensions were used for this study? For better clarity Figure 2 should have a legend. “The metrics the two methods” -> “the metric which the two methods” ?

Reviewer 3



After the author response: I'm upvoting the paper. Looking forward to seeing experiment extensions in the future. Summary: The paper proposes a "global anchor method" for detecting corpus-level language shifts as an alternative to the more common "alignment method", both leveraging distributed word representations. A proof is provided for the theoretical equivalence of the two methods, but the advantages of the former are highlighted as it allows to straightforwardly compare word embeddings of different dimensionality, along with some downstream advantages in uncovering the finer details of language shift. In my view, this is a very nice, albeit not well-rounded contribution, where the method exposition gets slightly more prominence than the experiments. Still, I vote accept: it is a strong, important contribution to the body of work in uncovering language shift and evolution. Strengths: - Very clear exposition, especially the motivation and the method equivalence proof. - The method is clean and simple, yet expressive. - The experiments are for the most part convincing. Weaknesses: - With limited space, the paper leaves quite some to wish for in terms of demonstrating the method application in actually uncovering (sets of interesting examples of) linguistic shift. Related work does a much better job at that, and the appendix of this draft does not contribute in this direction. A lot of space is dedicated to method equivalence, which is admirable, but it also shifts the weight of the contribution towards the first part of the writeup. - Along similar lines, only Google Books and arXiv are used, while related work does more. Altogether this indicates a slightly unbalanced contribution, where the empirical work slightly suffers. Questions: - Isn't the treatment of the "alignment method" slightly unfair with respect to dimensionality? If the corpora are available for building the embeddings, it seems a triviality to make the embeddings dimensions uniform.